# Norms for 150 consumer products: Perceived complexity, quality objectivity, material/experiential nature, perceived price, familiarity and attitude

**Filipe Loureiro**[1]*, **Teresa Garcia-Marques**[1], **Duane T. Wegener**[2]

**1** William James Center for Research, ISPA- Instituto Universitário, Lisboa, Portugal, **2** The Ohio State University, Columbus, Ohio, United States of America

* floureiro@ispa.pt

**Data Availability Statement:** Data and supplementary materials are available at https://osf.io/j5v8p/.

## Abstract

Consumer products are widely used as stimuli across several research fields. The use of consumer products as experimental stimuli lacks, however, the support of normative data regarding product features variability. In this work, we provide a first set of norms for people's perceptions of 150 consumer products regarding six relevant dimensions: product perceived complexity, quality objectivity, material/experiential nature, perceived price, familiarity and attitude. Products available in this normative database showed good overall distribution across the range of the dimensions evaluated. Obtained correlations between some of these dimensions provided evidence of how they can be confounded across products, further justifying the need to control for these dimensions. These norms should aid future research by allowing researchers to select products according to specific attributes and achieve appropriate experimental control. The norms here provided should also aid consumer behavior practitioners (such as marketers and advertisers) by providing insights as to how consumers perceive products along relevant dimensions.

## Introduction

Consumer products are not only to a great extent present in our daily lives, they have also been the focus of much research conducted on consumer behavior. Typically, research has addressed the characteristics that make products more or less appealing as well as how these characteristics influence consumers' lives. Thus, researchers have focused on factors such as consumer products' country-of-origin (e.g., [1,2]), product consumption as a measure of personal uniqueness (such as the possession of scarce products: e.g., [3–5]; consumer innovativeness: e.g., [6,7]; or product customization: e.g., [7], new product adoption (e.g., [8,9] and purchase intentions (e.g., [10]), among others.

Because consumer products are something to which participants can easily relate, these have also been extensively used as experimental stimuli across other research areas: judgment and decision-making (e.g., [11–13]), mere exposure effects (e.g., [14–16]), attitudes and persuasion (e.g., [17,18]), eye-tracking (e.g., [19,20]), decision preferences (e.g., [21,22]), and

**Funding:** This work was supported by FCT (Fundação para a Ciência e a Tecnologia) under grants SFRH/BD/110316/2015 awarded to Filipe Loureiro and UIDB/04810/2020.

**Competing interests:** The authors have declared that no competing interests exist.

memory (e.g., [23,24]; see [25]), among others. Researchers from these fields have the need to control their experimental stimuli in order to guarantee the validity of their operationalizations. Such control can be attained either by extensive pretesting of materials or reliance on general norms published in the literature (i.e., norms for people's shared perceptions of features specific to particular stimuli).

Our goal here is to share with the scientific community a normative dataset for people's perceptions of several consumer products regarding relevant dimensions. This work should contribute to a faster development of appropriate operationalizations that correspond to conceptual variables addressed by a number of theoretical approaches in the extant literature (e.g., theories of consumer perception, choice and behavior). We thus aim to address the operational gap between theories to be tested and the stimuli that might be used to address the conceptual variables addressed by the theories.

Such a goal arises from the fact that the widespread use of consumer products as experimental stimuli lacks the support of normative data that organizes and presents a set of relevant product features.

Our approach in the present work is the same followed in many research areas, wherein norms have been developed to allow researchers to select stimuli according to specific attributes in order to achieve appropriate experimental control (see [26]). The present normative database will support future research efforts to avoid the interference of confounding factors concerning consumer product dimensions, allowing researchers to select product stimuli in a way that controls for these features.

Besides providing a valuable research resource, these consumer product norms may also offer consumer behavior practitioners (such as marketers and advertisers) insightful information as to how consumers perceive products along relevant dimensions.

The selection of the consumer product dimensions for this normative study was tied to relevant research and theory in the consumer literature. For instance, we rely on research suggesting that people's perceptions of product features, such as perceived product complexity, quality objectivity, and relation to material/experiential purchase goals have a strong impact on their purchase decision-making styles (e.g., [21,22,27 (manuscript in preparation)]). Bellow, we review each of these dimensions regarding their relevance and operationalizations across the existing literature.

We first define what "consumer products" are and then provide an overview of the literature that informs researchers and practitioners about the relevant dimensions along which people tend to evaluate these products.

## Consumer products as multiple dimensional percepts

Consumer products are defined as products and services bought by final consumers for their personal use [28,29]. Consumer products include physical objects that can be offered for acquisition, use and consumption that might satisfy a want or a need. Services are products that consist of activities, benefits, or satisfactions that are essentially intangible [29]. Consumer products can be PCs, foods, cars, etc., and services include hotel stays, experiences, banking, insurance, etc.

These products are multidimensional percepts, and each of their dimensions likely influences how consumers relate with them. We first review some of the consumer product dimensions that have been the focus of research attention (see Fig 1) before operationalizing them in our normative study.

**1. Product complexity.** Perceived complexity has been methodologically operationalized in terms of the number of attributes that compose a product. For instance, products have been

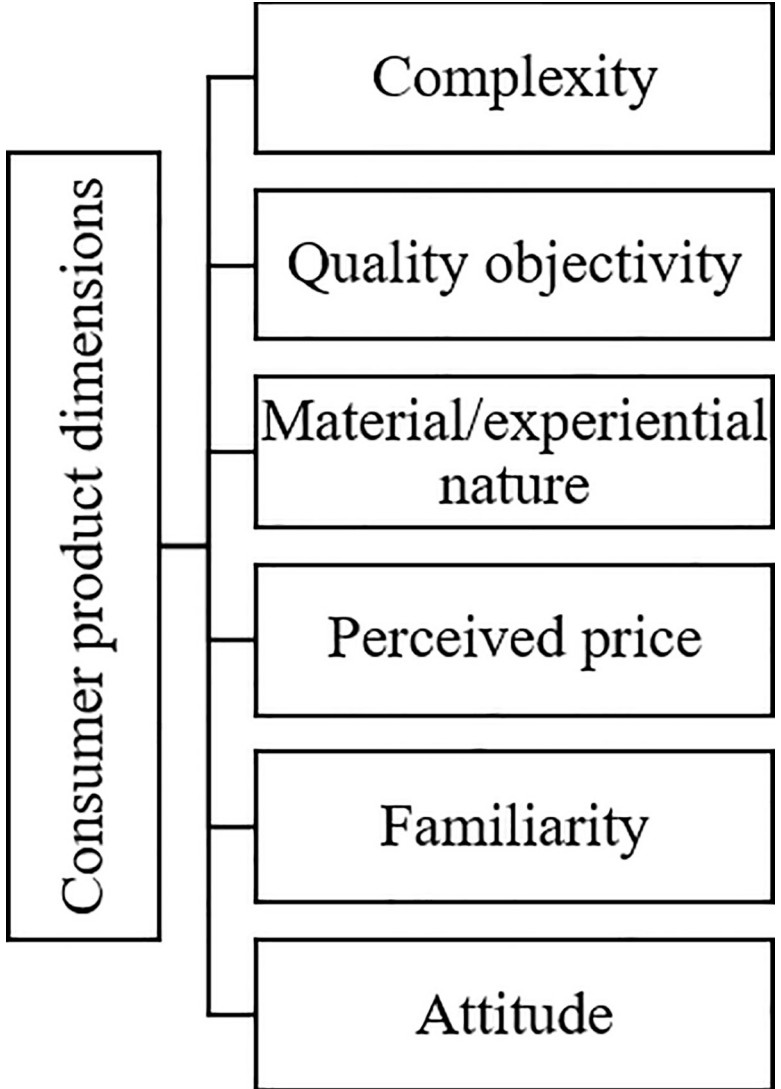

**Fig 1. Consumer product dimensions.**

described as complex when they are characterized by a large number of attributes that are relevant for the purchase decision [30]. Similarly, Netzer and Srinivasan [31] described products and services as complex when they are composed by ten or more attributes. According to Dijksterhuis ([32]; see also [33,34]), complexity is defined by the amount of information and facets a choice entails, meaning that a choice between products for which many attributes are important is complex, whereas a choice between products for which few attributes are important is simple.

Researchers have measured product complexity by assessing the number of attributes that can objectively characterize a product (e.g., [31,35,36]), by asking participants how many aspects of the product they would take into account when making a purchase decision [34] or by asking participants how complex they perceived the product on a "simple/not complex– complex" scale (e.g., [37–39]). Researchers have manipulated complexity by manipulating the number of attributes describing a product (e.g., [34,40]).

Product complexity is an important dimension of consumer products. Research on product complexity has provided evidence that: people are more willing to read instructions about

more complex products (e.g., [38]) and to actually read the instructions longer [41] when the product is relatively complex rather than simple. Consumers also prefer to choose complex products on the basis of rational analysis and simple products on the basis of intuition [22,27 (manuscript in preparation)], but are more satisfied with their purchases after choosing complex products intuitively and simple products conscientiously [32,34,42].

**2. Quality objectivity.**   Product quality has been theoretically and empirically defined in many different ways in the literature. Some definitions focus on product quality as something measurable and usually expressed by measurable product features (e.g., [43,44]). Other definitions focus on *consumers' perceptions* of quality, defining it as the consumer's judgment about a product's overall excellence and superiority (e.g., [45–48]), the customer's perception of the overall quality of a product, with respect to its intended purpose, in relation to alternatives [49], or as the degree to which a product or service fits the customer's needs and expectations (e.g., [50,51]).

For the purposes of the norms here presented, we focus on people's *perceptions* of how a product's quality can be evaluated on the basis of *objective* versus *subjective* dimensions. Whereas *objective product quality* refers to the product's actual performance, reliability, durability and serviceability–that is, objective facts and data—(e.g., [52,53]), *subjective product quality* is reflected by consumers perceptions of subjective attributes and personal tastes, opinions and preferences [48,54–56].

To the best of our knowledge, only one study, conducted by Inbar and colleagues [22], has measured participants' perceptions of choice quality objectivity. Specifically, in this study participants were asked to rate 25 choices in terms of the extent to which evaluation of the outcome was an objective or a subjective matter. Despite the fact that little is known about the impact of people's perceptions of choice quality objectivity, existing evidence supports the importance of controlling for such a dimension. Specifically, Inbar and colleagues [22] found that choices with objectively evaluable outcomes led participants to prefer to make their decisions in a rational way, whereas choices with subjectively evaluable outcomes led participants to prefer to make their decisions based on their intuitions. This result has important implications for what kinds of advertisements or information might be effective in advocating purchases of particular kinds of products and for what kinds of settings might enhance versus detract from the effectiveness of such influence attempts.

**3. Material versus experiential nature of products and purchases.**   A substantial amount of empirical work has focused on distinguishing material from experiential products and purchases. Van Boven and Gilovich [57] theoretically defined material products as tangible and material purchases as aimed at acquiring a product that one will keep in their possession. That is, material purchases involve products that one acquires with the intention of obtaining and having a physical good. Examples of material goods include cars, houses, and furniture. In contrast, experiential products are not tangible, and experiential purchases are made with the primary intention of acquiring an experience–an event through which one lives. Purchases of concert tickets, dining at restaurants, taking vacations, and visiting amusement parks are examples of experiential purchases.

The material-experiential distinction can be represented in a continuum by relying on consumers' personal intentions and motivations for the purchase [57]. For some purchases, delineating a distinction between experiences and material possessions may be difficult, but research suggests that participants and judges alike are able to identify the differences in these categories and reliably categorize purchases as material or experiential (e.g., [57,58]) as well as rate them on the material-experiential continuum [59,60]. Further justifying the importance of controlling for the material-experiential dimensions in consumer products and similar to results obtained regarding perceived product complexity and people's preferences for intuition

and rationality [22], research has shown that people tend to weight intuition more heavily with regards to experiential purchases and weight deliberation more heavily when making material purchases [21].

Documenting the relevance of this dimension, research on the distinction between material and experiential purchases has shown, among other findings, that experiential purchases make people happier than material purchases (e.g., [57]), experiences tend to be more closely associated with the self than possessions [61], and the evaluation of experiences is less comparative than that of possessions [58]. Also, whereas material purchase decisions are more likely to lead to buyer's remorse, experiential purchase decisions are more likely to lead to regrets of missed opportunities [62].

**4. Perceived price.**   The price of a product represents the amount of expenditure in a purchase transaction [63]. According to Jacoby and Olson [64], price can be categorized into objective and perceived price. Whereas objective price corresponds to a product's actual monetary cost, perceived price is defined as the consumer's subjective perceptions [64] and feelings [48] regarding the price of a product. Perceived price has also been defined as what the consumer sacrifices in order to obtain a product or service [48,65–67].

Perceived price relates to consumers' judgments of performance (e.g., [48]) and judgments of product quality (e.g., [68,69]). Perceived price fairness positively influences consumer trust [70], purchase decisions (e.g., [70,71]) and repurchase intentions (e.g., [72,73]). Perceived price also moderates the relation between quality of food and customer satisfaction [74]. Finally, increases in the perceived price of drinks increases subjective reports and neurological (fMRI) evidence of flavor pleasantness [75].

Consumers often compare the objective price with an overall prince range they perceive for the product category [76]. Research shows that consumers do not always know or remember the objective price of a product or service. Rather, they encode the price in ways that are meaningful to them [77]. Hence, consumers tend to remember the price of a product as "cheap" or "expensive" rather than as the dollar amount [78]. Accordingly, researchers have measured perceived price simply by asking participants to assess how inexpensive-expensive (e.g., [78–81]) or pricey-not pricey (e.g., [79,81]) products are.

Perceived price is also a relevant variable to control in research due to its intrinsic association with product perceived complexity. The more a product is perceived as relatively complex the higher its perceived price [22]. Consequently, when manipulating perceived complexity, unless precautions are taken, researchers are also manipulating perceived price.

**5. Product familiarity.**   Product familiarity is defined as the level of previous direct and indirect usage experience accumulated by the consumer (e.g., [82,83]). Researchers have measured product familiarity by asking participants how familiar-unfamiliar they are with a given product (e.g., [84–86]) or the features of that product [85,87].

A large amount of work has provided evidence of how product familiarity influences the way consumers process information and make decisions. For instance, product familiarity influences search for product information, depth of processing of such information, and choice confidence in decision-making [82,88]. More specifically, higher levels of product familiarity lead to the simplification of information processing through the use of nonfunctional cues (such as country of origin, brand, price) as heuristics to infer intrinsic product attributes, leading to more confidence in and reliance on such cues [89,90]. Product familiarity is also negatively associated with willingness to look for and read warnings [91,92] and positively associated with purchasing behavior (e.g., [93]).

Another dimension closely related to product familiarity is purchase frequency. In fact, research has combined measures of how familiar people are with certain products along with how frequently they buy these products in order to create a product familiarity index (e.g.,

[84,86]). Despite being associated, the use of both measures as an index of product familiarity might be problematic for some products (e.g., one might be extremely familiar with razor blades and its features but only so often purchase such a product). To that extent, in the present examinations, we measured both product familiarity and purchase frequency and separately present the norms for both dimensions.

**6. Product attitude.**   The term attitude refers to an overall evaluation of a particular target, such as people, issues and objects (e.g., [94,95]). Accordingly, product attitude has been generally defined as an overall evaluation of a particular product in a favorable or unfavorable manner (e.g., [96]). Researchers have measured product attitude by asking participants about how much they like the product, feel positive/negative towards it (e.g., [97–100]), and how good/bad and desirable/undesirable (e.g., [97,100]) the product is.

Most research on product attitudes has focused on this variable as an outcome. For instance, research has shown that product attitude is influenced by factors such as country of origin (e.g., [101]), packaging (e.g., [102]), tactile and visual inputs [103], peer communication [100], online reviews [104], and use of narrative online advertisement [105], among other findings. However, product attitude has also been identified as a predictor for relevant outcomes, such as purchase intentions (e.g., [100,106–109] and actual purchase behavior (e.g., [110]). Product attitude is clearly a key dimension of consumer products. When aiming to control for product features in research, researchers could greatly benefit from having an a priori indicator of how people evaluate different consumer products in a favorable or unfavorable manner.

## Development of consumer product norms

The above literature review demonstrates a number of relevant dimensions along which these products can be evaluated. Based on this evidence, we developed a descriptive study that aims to provide a set of norms for people's perceptions of a pool of 150 consumer products on each of these dimensions. The products in this normative study were selected with the goal of including products that are spread across much if not all of the rating range for each dimension (i.e., products perceived as high and low on each of the dimensions evaluated). We also analyze how the dimensions relate to each other across all products, allowing for a further analysis of how choices based on a single dimension can lead to confounds with other related dimensions.

## Method

### Participants

A sample of 389 North-American participants (48.3% females; $M_{age}$ = 37.24, $SD$ = 13.38) was obtained through online recruitment on Prolific Academic. All participants' native language was English, and they were living in the United States at the time of their participation.

### Ethics statement

This research was approved by the Research Ethics Committee at ISPA–Instituto Universitário, Lisbon, Portugal (approval number #D/006/10/2018). All participants provided consent by choosing to agree with the provided information about the study, prior to beginning the survey.

### Stimuli

A list of 150 consumer products (word stimuli) was assembled based on 1) products that had previously been classified along the perceived dimensions by previous research (e.g., complexity: [21,34]) and 2) products found on catalogs from several major store chains. The full list

can be consulted in the S1 File and includes products such as food, clothing, electronics, household products, experiences and services.

To prevent task fatigue and demotivation, each participant evaluated 54 products from the total list of 150 products for one of the six dimensions described. From those 54 products, participants first evaluated a set of 6 calibration products that spanned the range of the single dimension evaluated. These calibration items were presented first with the aim of providing participants a sense of the range of the stimuli to be evaluated on that dimension (e.g., [111–113]). After evaluating the calibration products, participants were presented with a random list of 48 additional products (From the total of 150 products, 6 calibration products were chosen to represent the range of values for that dimension and were used to start the ratings, leaving a total of 144 non-calibration products. Three lists of 48 products (144/3) were randomly created for each dimension. Participants were randomly assigned to one of these lists and one of the dimensions, and they evaluated the 48 products plus the 6 calibrators, resulting in a total of 54 products).

The calibration products for each dimension (presented in Table 1 and bolded in Tables 1–6 of the S1 File) were selected based on previous research and pre-testing (e.g., perceived complexity: [22,34]; quality objectivity: [22]; material/experiential purchase: [21,114]). In total, ratings were collected for 150 products, with each product rated by at least 20 participants (the 6 calibration products were rated by all participants). The average number of participant ratings per product was 23.3.

## Procedure and measures

We created an online survey using the Qualtrics survey platform. After providing informed consent to participate in the study, participants learned that the purpose of this research was to investigate people's perceptions of consumer products and experiences. Initial instructions indicated that their participation would involve rating a set of 54 consumer products and experiences with regard to a specific dimension.

After instructions, participants first evaluated the set of six calibration products, followed by ratings of 48 products on the same dimension. Product labels were individually displayed in a random order at the center of the screen with the dimension response scale presented on the same screen below the product label. Participants rated each product on all three items for the dimension before advancing to the next product. No time limits were imposed on responses, and participants were told that there were no correct or incorrect answers and that their personal opinion was of particular interest to the researchers. Participants rated each product by choosing the number that best corresponded to their evaluation of the product for the given dimension.

Before evaluating the products, participants were given a brief description of what the dimension being evaluated entailed. We provided this description with the aim of having all participants interpret the assessed dimensions the same way. These brief descriptions, as well as the response items used to evaluate the products on each dimension, are presented in Table 1. We based the descriptions and items on previous work and the literature reviewed. We used three items to measure each dimension for reliability purposes and in order to control for the possibility of these dimensions being multidimensional. By calculating the internal consistency of the three items for each dimension across the products we can examine how consistently the items tap into the proposed dimensions.

## Results

### Dimension evaluations

We started by analyzing participants evaluations regarding each of the 6 dimensions across all products. Table 2 presents the mean ratings across products for each dimension and the

**Table 1. Initial brief descriptions and items used for each dimension.**

| Dimension | Initial instruction | Calibration products | Items: |
|---|---|---|---|
| Perceived complexity | Some products/goods are relatively simple and have very few important aspects affecting their quality. These tend to be rather unidimensional. Other products/goods are more complex and have many core aspects affecting their quality. These tend to be relatively multidimensional. With this in mind, in this study, you will rate 54 consumer products regarding your own perceptions of complexity of these products. | High calibrators: • Car • Desktop computer • Room (renting) Low calibrators: • Umbrella • Dishwashing brush • Oven mitts | 1. How complex is this product? (1-Very simple; 6-Very complex) 2. How many aspects of this product could you take into account when making a purchase decision? (1–1 aspect; 2–2–3 aspects; 3–3–5 aspects; 4–5–7 aspects; 5–7–9 aspects; 6–10 or more aspects; based on [34]) 3. To what extent is this product relatively unidimensional or relatively multidimensional? (1-Relatively unidimensional; 6-Relatively multidimensional; based on [21]) |
| Quality objectivity | For some products/goods, one can objectively quantify whether their quality is good or bad. In these cases, the product's quality is based on facts. For other products/goods, whether a product's quality is good or bad is a subjective matter, depending on personal taste. In these cases, the quality of the product is merely a matter of opinion. With this in mind, in this study, you will rate 54 products regarding your own perceptions of how objective these products' qualities are. | High calibrators: • Paper clips • Hangers • Medical treatment Low calibrators • Vacation package • Dessert at a restaurant • Entrée at a restaurant | 1. To what extent is the evaluation of this product's qualities a subjective or an objective matter? (1-Mainly a subjective matter; 6-Mainly an objective matter; based on [22]) 2. To what extent does the quality of this product depend on a personal taste or is objectively the same for everyone? (1-Quality depends on personal taste; 6-Quality is objective and the same for everyone) 3. To what extent is the quality of this product a matter of opinion or a function of facts? (1-Quality is a matter of opinion; 6-Quality is a function of facts) |
| Material/ experiential purchase | Some purchases are material, tangible and purchased with the intention of acquiring and having a physical good. Other purchases are experiential, reflecting events that are lived through and purchased with the intention of acquiring experiences (i.e., with doing something). With this in mind, in this study, you will rate 54 products regarding your own perceptions of these products' experiential or materialistic characteristics. | High calibrators: • Vacation package • Museum ticket • Dinner at a restaurant Low calibrators: • Suit • Necklace • Vase | 1. To what extent is the purchase of this product a material purchase or an experiential purchase? (1-Definitely material; 6-Definitely experiential; based on [115]) 2. To what extent does the purchase of this product emphasize possession of an object or experiencing an activity? (1-Definitely emphasis on possession; 6-Definitely emphasis on experiencing) 3. To what extent is the purchase of this product focused on having or focused on doing? (1-Definitely focused on having; 6-Definitely focused on doing) |
| Perceived Price | A product's price is the amount of expenses incurred in purchase transactions. While a product's objective price represents the actual price of the product, perceived price is the subjective perception people have of the objective price of a product. While people do not always remember the exact price of a specific product or service, they may remember the price as relatively "cheap" or "expensive". With this in mind, in this study, you will rate 54 products regarding your own subjective perceptions of these products' prices. | High calibrators: • House • Car • Cruise trip Low calibrators: • Dishwashing brush • Pen • Paper clips | 1. How expensive is this product? (1-Very inexpensive; 6-Very expensive; (based on [78–81]) 2. How pricey is this product? (1-Not pricey at all; 6-Very pricey; based on [79,81]) 3. How high is this product's price? 1-Very low; 6-Very high; based on [79,81]) |
| Familiarity | People may have different levels of familiarity with different products/ goods. For instance, a person may have a lot of prior experience with a type of product or purchase it very frequently. Conversely, a person may be less familiar with a given product or its features, or buy it less frequently. With this in mind, in this study, you will rate 54 consumer products regarding your own levels of familiarity with these products. | High calibrators: • Shampoo • Breakfast cereal • Chewing gum Low calibrators: • Life insurance • Cruise trip • 3D printer | 1. How familiar are you with this product? (1-Not at all familiar; 6-Extremely familiar; based on [84–86]) 2. How familiar are you with the features of this product? (1-Not at all familiar; 6-Extremely familiar; based on [85,87]) 3. How frequently do you buy this product? (1-Never; 6-Very frequently; based on [84,86]) |
| Attitude | People have different attitudes and feelings towards different products and services. On this basis, in this study, you will rate 54 products regarding your own evaluations of how much you like these products. | High calibrators: • Ice cream • Vacation package • Massage Low calibrators: • Toilet brush • Insecticide • Cigarettes | 1. How positive do you feel about this product? (1-Not positive at all; 6-Very positive; based on [98,99]) 2. How negative do you feel about this product? (reverse-coded) (1-Not negative at all; 6-Very negative; [98,99]) 3. How much do you like this product? (1-I don´t like this product at all; 6-I like this product very much; based on [98,99]) |

**Table 2. Descriptive statistics of mean dimension ratings.**

| Dimension | M | SD | Average Alpha | Low calibrators' M | High calibrators' M |
|---|---|---|---|---|---|
| Perceived complexity | 2.80 | 1.00 | 0.84 | 1.91 | 4.79 |
| Quality objectivity | 3.55 | 1.14 | 0.85 | 2.65 | 4.52 |
| Material/experiential nature | 2.97 | 1.36 | 0.88 | 1.90 | 5.15 |
| Perceived price | 2.58 | 0.97 | 0.94 | 1.76 | 5.12 |
| Familiarity | 4.45 | 1.02 | 0.60 | 3.01 | 5.13 |
| Attitude | 4.67 | 1.08 | 0.79 | 3.03 | 4.94 |

respective standard deviations. The table also presents the average internal consistency of the three items used for each dimension across products, and the average means for the low and high calibration products used for each dimension.

Average Cronbach alphas suggested good internal consistency of the items used to evaluate each dimension. The internal consistency of items used to evaluate product familiarity was lower in comparison to the other dimensions, however, suggesting that product familiarity and purchase frequency might reflect different dimensions. The mean value for each of the three items is presented individually for all products in the S1 File (Tables 1–6 in S1 File), not only for the familiarity/acquisition frequency dimension but also for the five other dimensions. This allows researchers not only to make use of the average of the three items but also to use of the mean values for each item.

By observing the average means in Table 2 for the low and high calibration products used for each dimension, we see that the calibration items fulfilled their goal of providing participants a sense of the range of the dimension evaluated, by being rated as considerably lower and higher, respectively, in comparison to the average means across all products.

In Table 3, we present a list of the 5 products with the most extreme positive and negative mean ratings on each dimension. These average ratings show a good distribution across the range in their different dimensions. Fig 2 presents the frequency distributions of ratings for all products for each of the six dimensions. These further confirm that the products show an overall variation across the whole range of the dimensions evaluated, with the exception of the attitude dimension, for which most products were rated positively (an issue with potentially negative products is that they do not remain available for long–or never make it to market–so using existing products might always skew toward mostly positive attitudes). The distributions in Fig 2 allow us to conclude that, for each dimension, we largely fulfilled our aim of obtaining products that were perceived across the whole range of the dimensions evaluated.

## Correlations between dimensions

Next, we computed correlations among the six dimensions across the evaluations of all 150 products, using products as the unit of analysis (i.e., correlating the mean values for each product's features). Overall, the results showed significant correlations between the dimensions (see Table 4). The stronger correlations show that: 1) perceived complexity correlated positively with perceived higher price; 2) the more products were perceived as experiential the less their quality was perceived as an objective matter; and 3) product familiarity correlated negatively with product perceived complexity and price and positively with product attitude favorability.

## Discussion

In this paper, we provide norms for people's perceptions of 150 consumer products on six relevant dimensions: perceived complexity, quality objectivity, material/experiential nature,

**Table 3. Products with the most extreme mean ratings per dimension (mean ratings in brackets).**

| Perceived complexity | Quality objectivity | Mat/exp. nature | Perceived price | Familiarity | Attitude |
| --- | --- | --- | --- | --- | --- |
| Car (5.12) | Trash bin (4.98) | Massage (5.64) | House (5.48) | Toilet paper (5.62) | Television (5.36) |
| House (5.12) | Aspirin (4.97) | Cruise trip (5.56) | Car (5.11) | Bread (5.61) | Pillow (5.30) |
| Convertible laptop (5.10) | Trash bag (4.92) | Vacation package (5.40) | Medical treatment (4.90) | Pizza (5.58) | Soap (5.25) |
| Laptop (5.08) | Scissors (4.92) | Museum ticket (5.27) | Cruise trip (4.76) | Soap (5.54) | Desktop (5.25) |
| Smartphone (4.98) | DVD player (4.82) | Concert ticket (5.22) | Vacation package (4.73) | Toothpaste (5.43) | Air conditioner (5.24) |
| Trash bags (1.52) | Breakfast cereal (2.02) | Laptop bag (1.70) | Postcard (1.40) | Motorcycle (3.06) | Motorcycle (3.57) |
| Cleaning cloth (1.45) | Ice cream (1.97) | Table (1.70) | Yogurt (1.33) | Cruise trip (2.81) | Soft drink (3.49) |
| Coasters (1.45) | Whiskey (1.97) | Underwear (1.59) | Soap (1.32) | Nature park ticket (2.80) | Disposable plastic cups (3.35) |
| Hangers (1.43) | Tea (1.97) | Wall clock (1.53) | Air freshener (1.32) | 3D printer (2.56) | Insecticide (3.22) |
| Toilet brush (1.38) | Chocolate bar (1.80) | Trash bin (1.52) | Chewing gum (1.25) | Power bank (2.41) | Cigarettes (2.07) |

perceived price, familiarity and attitude. To our knowledge, this the first available consumer products normative database assessing these dimensions.

The products in this normative database showed a good overall distribution across the rating range of the dimensions evaluated. We thus fulfilled our secondary aim of obtaining products perceived as relatively high and low on these dimensions.

Correlations between dimensions across all products replicate evidence from the previous literature where relevant pairs of dimensions have been examined (e.g., [22], for the perceived complexity-price association; [56], for the experientiality-quality objectivity association). These results provide further evidence that using one dimension to choose stimuli can create a set of stimuli that confounds the dimension used to choose the stimuli with one or more other product dimensions, further justifying the need to control for these alternative dimensions.

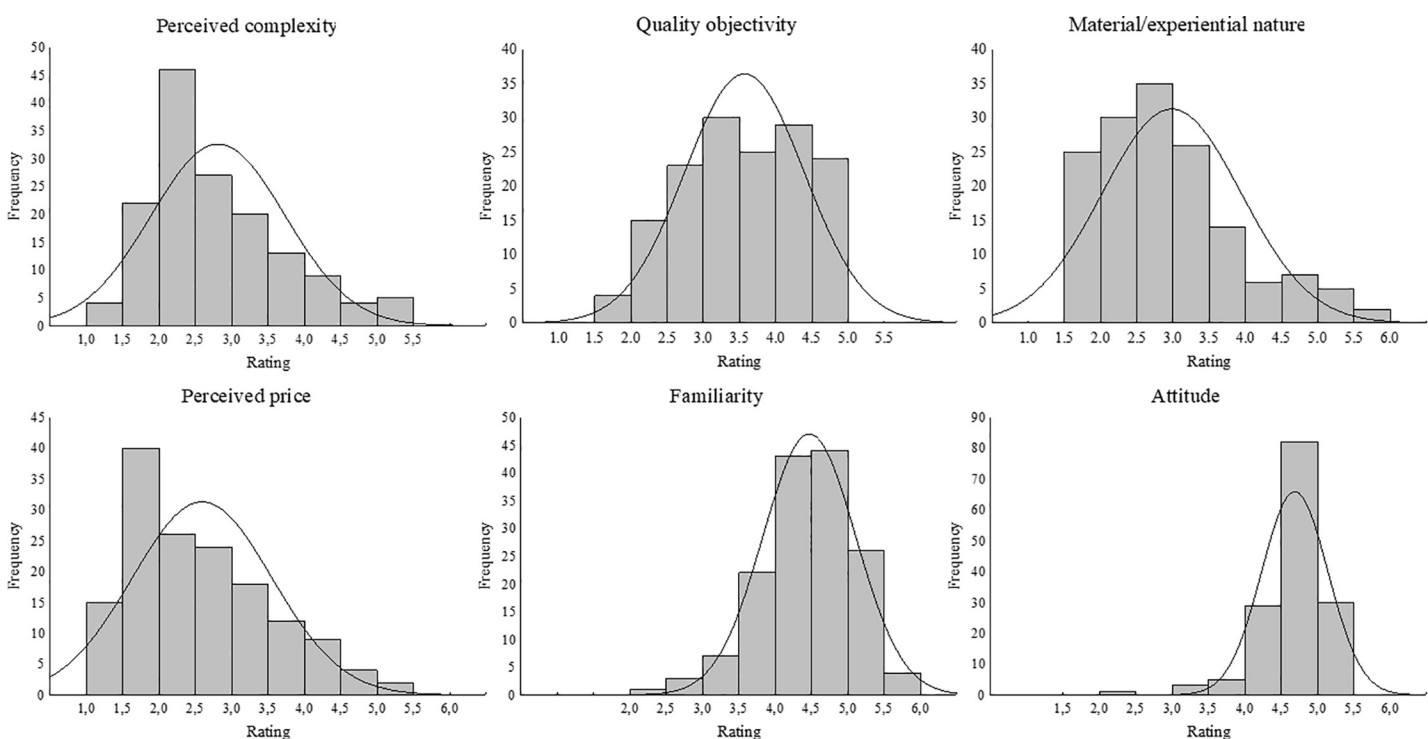

**Fig 2. Distribution of ratings across products for each of the 6 dimensions.**

**Table 4. Correlations between dimensions across all products.**

| Dimensions | 1 | 2 | 3 | 4 | 5 | 6 |
|---|---|---|---|---|---|---|
| 1) Perceived complexity | - | | | | | |
| 2) Quality objectivity | .05 | - | | | | |
| 3) Material/Experiential nature | .18* | -.39** | - | | | |
| 4) Perceived price | .86** | .03 | .13 | - | | |
| 5) Familiarity | -.44** | -.10 | -.16* | -.55** | - | |
| 6) Attitude | .17* | -.10 | -.02 | .10 | .31** | - |

*. Correlation is significant at the 0.05 level (2-tailed)

**. Correlation is significant at the 0.01 level (2-tailed).

The norms here presented were obtained with North-American participants. As with other normative databases, generalizations to other populations and cultures should be made with caution and cross-validation is recommended. For instance, product familiarity is an important dimension to take into account in cross-cultural studies. For example, food products can be differently perceived depending on the consumers' western and eastern cultural backgrounds (e.g., [116,117]). Similarly, differences in perceived prices may be observed when examining products across different countries and cultures. Therefore, future research should consider extending these norms to other countries/cultures.

Additionally, we did not measure individual differences in participants' consumption behaviors or preferences. For example, individual differences in preferences for more material items or life experiences [118], green purchasing behavior (e.g., [119]) and hedonic/utilitarian purchase motivations (e.g., [120,121]) might influence consumers' perceptions of the dimensions here assessed, and future researchers might consider controlling for such individual differences if extending or applying these norms.

Overall, the norms here presented constitute a valuable resource that will allow researchers to select consumer products according to specific attributes and facilitate researchers' choices aimed at achieving appropriate experimental control. That is, using the dimension values as independent variables can often be done while avoiding confounds with alternative dimensions (or at least being aware of some related dimensions that could inform interpretation of reactions to the chosen stimuli).

To this extent, in addition to the S1 File providing descriptive statistics for all 150 consumer products for each dimension, a user-friendly spreadsheet is made available (see Data Availability section), that allows researchers to sort (in ascending or descending order) all products for a specific dimension, allowing for a quick examination of the product's characteristics across all the dimensions.

These norms should also aid consumer behavior practitioners in the sense that they provide insightful information as to how consumers perceive products on a variety of relevant dimensions.

## Supporting information

**S1 File.**
(DOCX)

## Author Contributions

**Conceptualization:** Filipe Loureiro, Teresa Garcia-Marques, Duane T. Wegener.

**Formal analysis:** Filipe Loureiro.

**Investigation:** Filipe Loureiro.

**Methodology:** Filipe Loureiro, Teresa Garcia-Marques, Duane T. Wegener.

**Supervision:** Teresa Garcia-Marques, Duane T. Wegener.

**Writing – original draft:** Filipe Loureiro.

**Writing – review & editing:** Filipe Loureiro, Teresa Garcia-Marques, Duane T. Wegener.

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
