## [Decision Letter · Decision Letter 0]

13 Mar 2020

PONE-D-20-00355

Norms for 150 consumer products: perceived complexity, quality objectivity, material/experiential nature, perceived price, familiarity and attitude.

PLOS ONE

Dear Mr. Loureiro,

Thank you for submitting your manuscript to PLOS ONE. After careful consideration, we feel that it has merit but does not fully meet PLOS ONE’s publication criteria as it currently stands. Therefore, we invite you to submit a revised version of the manuscript that addresses the points raised during the review process.

The two reviewers have a very different view about the paper. Reviewer #1 provides a sound and logical reasoning around the main weakness of the paper and useful suggestions to improve it (please consider the last recommendation of Reviewer #1 as as suggestion for future research as, of course, it would mean to write a different paper). I believe you should carefully consider the points related to the theoretical contribution and the lack of a clear theoretical framework and related hypotheses. Reviewer #2 liked he paper, but this is basically all what we know as a very limited reasoning is supporting the provided statements. 

We would appreciate receiving your revised manuscript by Apr 27 2020 11:59PM. To enhance the reproducibility of your results, we recommend that if applicable you deposit your laboratory protocols in protocols.io, where a protocol can be assigned its own identifier (DOI) such that it can be cited independently in the future. For instructions see: http://journals.plos.org/plosone/s/submission-guidelines#loc-laboratory-protocols

We look forward to receiving your revised manuscript.

Kind regards,

Claudio Soregaroli

Academic Editor

PLOS ONE

Journal Requirements:

ISPA's Research Ethics Committeee approved this study, through writen consent

(#D/006/10/2018).

Reviewers' comments:

Reviewer's Responses to Questions

**Comments to the Author**

1. Is the manuscript technically sound, and do the data support the conclusions?

Reviewer #1: Partly

Reviewer #2: Yes

2. Has the statistical analysis been performed appropriately and rigorously? 

Reviewer #1: Yes

Reviewer #2: Yes

3. Have the authors made all data underlying the findings in their manuscript fully available?

Reviewer #1: Yes

Reviewer #2: Yes

4. Is the manuscript presented in an intelligible fashion and written in standard English?

Reviewer #1: Yes

Reviewer #2: Yes

5. Review Comments to the Author

Reviewer #1: In principle, I like and support the idea of studying relevant dimensions for consumer products. But right now, based on the introduction, I do not see any theoretical justification for conducting this study. For example, what are the theoretical gaps the authors are trying to address? This is important to gauge the contribution of their work.

Your introduction introduces too many concepts and can be distracting from the main theme. It would also help to define constructs (albeit briefly) like “complexity”, ‘quality objectivity” etc. As researchers you may be familiar with these, but readers may not be. Further, I would like to know what theoretical gaps, specific to your variables of interest currently remain unaddressed.

I think the literature review is very simplistic. Consumer behaviour and product decisions can be nuanced, rather than being based on ‘rational’ versus ‘intuition’ dimension. The research should focus on consumer variables (e.g., individual differences), that drives the perception of product differences.

Right now, the research lacks a conceptual model. This is tied to my previous point that understanding perceptions of differences based on consumer related variables will make interesting contribution. Right now, all we know that significant differences exist between products across different dimensions. This is quite intuitive and already known.

I think the study should gain from research hypotheses. I am not sure what are the hypotheses that researchers are trying to study. Having a hypothesis followed by appropriate empirical analysis would give me more confidence in the scientific nature of findings.

I find some of the constructs (e.g. ‘attitude” ) are overly simplified. Measures for attitude are quite well established and consist of multiple items. Given the nature of their products, more items would have been used to capture this construct better.

I think the reliability for ‘familiarity’ is low. I would therefore doubt the validity of the findings, as to me, the scale items for this construct do not hold together.

I think the respondents have to answer at least 48X3= 144 items to complete their survey. It is quite obvious that there will be fatigue. Can the researchers clarify, what they did to deal with this issue?

The discussion needs to be more fleshed out in terms of contribution. Right now, I cannot see any theoretical contribution, but a summary of discussion. If I know what authors claim to have discovered, will it improve the research process (e.g., experimental design etc.). Please let the readers know.

Please conduct an additional study with relevant product dimensions but this time around with individual difference variables. If possible, run this study in a different culture. I would like to see the results being replicated, and also driven by consumer differences like the authors have posited in their theory.

Good luck with your work!

Reviewer #2: The paper deals with a very interesting topic, it was well written and it was pleasant to read.

The introduction section well describes all the relevant literature, no important papers are missing. Overall, the research has been well conducted and the design of the experiment is almost flawless. Most importantly, the results provide a very strong evidence that choosing one dimension can create a set of confounding stimuli.

6. PLOS authors have the option to publish the peer review history of their article (what does this mean?). If published, this will include your full peer review and any attached files.

Reviewer #1: No

Reviewer #2: No

---

## [Author Response · Author response to Decision Letter 0]

28 Apr 2020

The text below is a direct copy of the content in the uploaded file "Response to reviewers":

Dr. Claudio Soregaroli

Academic Editor

PLOS ONE

Dear Dr. Claudio Soregaroli,

First of all, we would like to thank you and the reviewers for the review process regarding our submission of the paper entitled “Norms for 150 consumer products: perceived complexity, quality objectivity, material/experiential nature, perceived price, familiarity and attitude.”

We appreciate the constructive criticisms and feedback received, and were happy that you and both reviewers see the potential of the submitted work. We appreciate, too, your selection of the main questions of the reviewers that are of your concern and thank you for inviting us to submit a revised version of the manuscript, which addresses the points raised during the review process.

We found it interesting that the two reviewers have a very different view about the paper, since it allows us to understand how the general public will receive our paper.

Below, we address Reviewer 1’s concerns, since he/she is the one that is more critical of it, pointing out what he/she understands as weaknesses of the paper.

As suggested by you, we carefully consider the points related to the theoretical contribution of the paper and the lack of a clear theoretical framework and related hypotheses. To this extent, we try to address all the concerns raised by Reviewer 1 in order to make it more clear what we aim with this paper.

We look forward to receiving your response to the new version of our revised manuscript and we thank you for your consideration and support to this work.

Kind regards, 

Filipe Loureiro

Teresa Garcia-Marques

Duane Wegener

Reviewers' comments and Authors responses

Reviewer #1

Comment 1: In principle, I like and support the idea of studying relevant dimensions for consumer products. But right now, based on the introduction, I do not see any theoretical justification for conducting this study. For example, what are the theoretical gaps the authors are trying to address? This is important to gauge the contribution of their work.

Response: Investigators from several research fields (as exemplified in the paper’s introduction, p.3, lines 11-15) will benefit from having access to extensive normative data on various objective and subjective dimensions related to experimental stimuli. Our aim was to share with the scientific community results from our pre-testing of materials regarding a set of theoretically-relevant dimensions of consumer products (as exemplified by research testing whether, for instance, consumers engage in higher scrutiny of complex products [38, 41], or how the material-experiential distinction is operationalized in a continuum [57], or how product familiarity influences the way consumers process information and make decisions [82,88-93], among others). The selected dimensions are highly relevant for the operationalization of variables that need to be either addressed or controlled for in many studies in several research fields. As such, Reviewer 1 is correct in saying that we are not addressing any particular theoretical gap. Instead, we are addressing an operational gap between theories that relate to dimensions of complexity, quality, experiential nature, etc. and stimuli that researchers could profitably use to study those dimensions.

We understand, with the help of the reviewer, that we were not clear or convincing presenting our goals in the first version of the paper. As such, we added some clarity to the first pages of the paper aiming to clarify our present goal as addressing the operational gap between theories to be tested and the stimuli that might be used to address the conceptual variables addressed by the theories. The conceptual variables are addressed through the presentation of norms concerning six product dimensions: product perceived complexity, quality objectivity, material/experiential nature, perceived price, familiarity and attitude. Although such changes were made across all the introduction, we highlight: 

Page 3, lines 15-23:

“Researchers from these fields have the need to control their experimental stimuli in order to guarantee the validity of their operationalizations. Such control can be attained either by extensive pretesting or reliance on general norms published in the literature. Our goal here is to share with the scientific community results of a set of norms for people’s perceptions of several consumer products regarding relevant dimensions, allowing for a faster development of appropriate operationalizations that correspond to conceptual variables addressed by a number of theoretical approaches in the extant literature. We thus aim to address the operational gap between theories to be tested and the stimuli that might be used to address the conceptual variables addressed by the theories.”

Page 4, lines 4-12:

“Our approach in the present work is the same followed in many research areas, wherein norms have been developed to allow researchers to select stimuli according to specific attributes in order to achieve appropriate experimental control (see [26]). The present normative database will support future research that wishes to avoid the interference of confounding factors concerning consumer product dimensions, allowing researchers to select product stimuli in a way that controls for these features. 

Besides providing a valuable research resource, these consumer product norms may also offer consumer behavior practitioners (such as marketers and advertisers) insightful information as to how consumers perceive products along relevant dimensions.”

Comment 2: Your introduction introduces too many concepts and can be distracting from the main theme. It would also help to define constructs (albeit briefly) like “complexity”, ‘quality objectivity” etc. As researchers you may be familiar with these, but readers may not be. Further, I would like to know what theoretical gaps, specific to your variables of interest currently remain unaddressed.

Response: Thank you for this comment. Based on it, we made an effort to be clearer with regards to the definition of the six dimensions and their relevance for the literature. Although changes were made across all the introduction, we highlight, with regards to this comment:

Page 4, lines 13-22:

“The selection of the consumer product dimensions for this normative study was tied to relevant research and theory in the consumer literature. For instance, we rely on research suggesting that people’s perceptions of product features, such as perceived product complexity, quality objectivity, and relation to material/experiential purchase goals have a strong impact on their purchase decision-making styles (e.g., [21,22,27 (manuscript in preparation)]). Bellow, we review each of these dimensions regarding their relevance and operationalizations across the existing literature. 

We first define what “consumer products” are and then provide an overview of the literature that informs researchers and consumers about the relevant dimensions along which people tend to evaluate these products.”

Comment 3: I think the literature review is very simplistic. Consumer behaviour and product decisions can be nuanced, rather than being based on ‘rational’ versus ‘intuition’ dimension. The research should focus on consumer variables (e.g., individual differences), that drives the perception of product differences.

Right now, the research lacks a conceptual model. This is tied to my previous point that understanding perceptions of differences based on consumer related variables will make interesting contribution. Right now, all we know that significant differences exist between products across different dimensions. This is quite intuitive and already known.

Response: In this comment, the reviewer emphasizes, once more, the concern regarding the lack of a theoretical framework to our approach. Although our aim is more methodological than theoretical, we understand the relevance of studying consumer variables (e.g., individual differences) that might lead, or not, to different perceptions of these dimensions across different products. We, too, believe that future work should follow-up on this question (and hence had already raised this point in the discussion section of our manuscript). However, we believe that such research pursuits would benefit from having norms across people along the relevant dimensions. Once those are known, one could conduct research aimed at examining how some consumer characteristic would influence the extent to which differences in one or more product dimension are perceived (or used in product evaluations). The reviewer might conceive of ways that such research could be conducted without starting with target products that normatively differ along the product dimensions of relevance to the consumer characteristics under study, but we believe that such pursuits might progress more smoothly and quickly if the researcher can start with a list of potential products that represent variations along the relevant dimensions (and at least some irrelevant dimensions that the researcher might want to control in order to be able to make claims specific to the dimension or dimensions of interest in the research). We hope that the current descriptions in the manuscript appropriately address the proposed goal of our paper – to provide a first set of norms for people’s perceptions of 150 consumer products – (even if this is not the one required to address the reviewer’s concerns). Instead of differentiating between individuals, we differentiate between consumer products (that could then be used as stimuli in future studies aimed at differentiating among individual consumers). 

Comment 4: I think the study should gain from research hypotheses. I am not sure what are the hypotheses that researchers are trying to study. Having a hypothesis followed by appropriate empirical analysis would give me more confidence in the scientific nature of findings.

Response: That approach is one we typically use in our own basic (theory-testing) research. However, as we now clarify in the manuscript, the goal of the current study is descriptive – i.e., to map out differences across products that might be used in future hypothesis-testing studies to either create relevant differences across stimuli or to control for differences that happen to be related to the key dimensions but are not intended to be the focus of the research. As such, this descriptive enterprise is not guided by hypotheses other than assuming that products vary along the dimensions of interest (to researchers and theories in consumer behavior, decision making, and related fields).

Please see page 12, lines 2-5:

“The above literature review demonstrates a number of relevant dimensions along which these products can be evaluated. Based on this evidence, we developed a descriptive study that aims to provide a set of norms for people’s perceptions of a pool of 150 consumer products on each of these dimensions.”

Comment 5: I find some of the constructs (e.g. ‘attitude”) are overly simplified. Measures for attitude are quite well established and consist of multiple items. Given the nature of their products, more items would have been used to capture this construct better.

I think the reliability for ‘familiarity’ is low. I would therefore doubt the validity of the findings, as to me, the scale items for this construct do not hold together.

Response: The reviewer’s concern about the validity and reliability of our measures are of great relevance. We acknowledge that our measures used relatively few items, but items that had been extensively used in previous research (as referenced on Table 1) and that corresponded well to the constructs they were intended to measure. In the case of product attitudes, these measurement items represented a number of the semantic differential dimensions routinely used since Osgood, Suci, and Tannenbaum (1957) and corresponding quite well to the evaluation of the product along the valenced dimension (bad-good or disfavor-favor) traditionally defining the attitude construct. All other constructs in our study were equally measured following approaches applied by previous research (as reviewed in the introduction and referenced on Table 1).

We understand and are sensitive to the reviewer’s concern about the measurement of product familiarity. By using as a reliability measure the Cronbach’s alpha, we should recognize that this indicator is highly sensitive to the number of items used to measure a construct. The formula for the Cronbach’s alpha coefficient implies that the greater the number of items present in a scale, the greater its reliability. Nevertheless, and because we agree with the reviewer in the essence of the comment regarding the construct of familiarity, we had already addressed this point in the introduction of our paper as well as in the results section:

Page 10, lines 18-2 (p.11):

“Another dimension closely related to product familiarity is purchase frequency. In fact, research has combined measures of how familiar people are with certain products along with how frequently they buy these products in order to create a product familiarity index (e.g., [84,86]). Despite being associated, the use of both measures as an index of product familiarity might be problematic for some products (e.g., one might be extremely familiar with razor blades and its features but only so often purchase such a product). To that extent, in the present examinations, we measured both product familiarity and purchase frequency and separately present the norms for both dimensions.”

Page 17, lines 21-28:

“The internal consistency of items used to evaluate product familiarity was lower in comparison to the other dimensions, however, suggesting that product familiarity and purchase frequency might reflect different dimensions. The mean value for each of the three items is presented individually for all products in the Supplementary Materials (Tables 1-6), not only for the familiarity/acquisition frequency dimension but also for the five other dimensions. This allows researchers not only to make use of the average of the three items but also to use of the mean values for each item.”

Comment 6: I think the respondents have to answer at least 48X3= 144 items to complete their survey. It is quite obvious that there will be fatigue. Can the researchers clarify, what they did to deal with this issue?

Response: Like the reviewer, we, too, were concerned about the participants’ potential task fatigue. This is the reason why we took several measures aiming to avoid fatigue, which we address in the manuscript. 

Page 12, lines 17-20:

“To prevent task fatigue and demotivation, each participant evaluated 54 products from the total list of 150 products, for one of the six dimensions described. From those 54 products, participants first evaluated a set of 6 calibration products that spanned the range of each of the six dimensions.”

Page 13, lines 20-1 (page 14):

“No time limits were imposed on responses, and participants were told that there were no correct or incorrect answers and that their personal opinion was of particular interest to the researchers.”

Additionally, since products were randomly ordered for each participant, it would also seem that the incidence of responses to a given product falling late in the presented list would be minimized to ensure that any fatigue-induced response tendencies would be equated across products and not represent the majority of responses for any target product. 

Comment 7: The discussion needs to be more fleshed out in terms of contribution. Right now, I cannot see any theoretical contribution, but a summary of discussion. If I know what authors claim to have discovered, will it improve the research process (e.g., experimental design etc.). Please let the readers know.

Response: In line with our response to the reviewer’s first comment, the goal of the work is methodological rather than theoretical. It will be potentially important, though, in that it will fill the important gap between theoretical attention to the dimensions of complexity, experiential nature, positivity (attitude), etc. and operationalizations of those concepts using specific target stimuli in empirical research (or control for dimensions a researcher wishes not to confound certain dimensions of interest). Specifically, the present normative data will allow researchers to overcome methodological limitations when using consumer products, and to select products according to specific attributes and achieve appropriate experimental control. 

To this extent, and in addition to the Supplementary Materials that provides descriptive statistics for all products according to each dimension (ordered in descending order), we will also make available a protected spreadsheet that will allow readers to sort all products according to the selected dimension, allowing for a quick examination of the product’s characteristics across all dimension, as exemplified below:

A paragraph was added to the discussion section regarding the availability of the tool described above:

Page 21, lines 13-17:

“To this extent, in addition to the Supplementary Materials providing descriptive statistics for all 150 consumer products for each dimension, a user-friendly spreadsheet is made available (see Data Availability section), that allows researchers to sort (in ascending or descending order) all products for a specific dimension, allowing for a quick examination of the product’s characteristics across all the dimensions.”

Comment 8: Please conduct an additional study with relevant product dimensions but this time around with individual difference variables. If possible, run this study in a different culture. I would like to see the results being replicated, and also driven by consumer differences like the authors have posited in their theory.

Response: As mentioned, even though we find such possibilities interesting and we believe that the current product norms would be helpful in selecting appropriate stimuli for studies of individual differences in reactions toward or use of the dimensions in product evaluations, these additional research questions go beyond our current goals with this paper. As such, we address them in the discussion section of the manuscript as new questions to be studied in future research.

Reviewer #2: 

The paper deals with a very interesting topic, it was well written and it was pleasant to read.

The introduction section well describes all the relevant literature, no important papers are missing. Overall, the research has been well conducted and the design of the experiment is almost flawless. Most importantly, the results provide a very strong evidence that choosing one dimension can create a set of confounding stimuli.

Response: We thank the reviewer for a clear understanding of our aims and the relevance of this research for the support of future research.

---

## [Decision Letter · Decision Letter 1]

21 May 2020

PONE-D-20-00355R1

Norms for 150 consumer products: perceived complexity, quality objectivity, material/experiential nature, perceived price, familiarity and attitude.

PLOS ONE

Dear Dr. Loureiro,

Thank you for submitting your manuscript to PLOS ONE. After careful consideration, we feel that it has merit but does not fully meet PLOS ONE’s publication criteria as it currently stands. Therefore, we invite you to submit a revised version of the manuscript that addresses the points raised during the review process.

Reviewer 1 still remains very critical about the article, highlighting a lack of contribution to theory and concerns regarding key constructs. Given the opposite views of the initial two reviewers, I decided to ask the opinion of a third reviewer. Reviewer 3 gives some chances to the article even if several critical aspects emerge that partially converge and reinforce the opinion of Reviewer 1. There is an important work to be done to make the article more clear to the reader, easy to follow, and better motivated. This effort could make the article also more convincing in terms of its contribution. 

We look forward to receiving your revised manuscript.

Kind regards,

Claudio Soregaroli

Academic Editor

PLOS ONE

Reviewers' comments:

Reviewer's Responses to Questions

**Comments to the Author**

1. If the authors have adequately addressed your comments raised in a previous round of review and you feel that this manuscript is now acceptable for publication, you may indicate that here to bypass the “Comments to the Author” section, enter your conflict of interest statement in the “Confidential to Editor” section, and submit your "Accept" recommendation.

Reviewer #1: (No Response)

Reviewer #3: (No Response)

2. Is the manuscript technically sound, and do the data support the conclusions?

Reviewer #1: Partly

Reviewer #3: Partly

3. Has the statistical analysis been performed appropriately and rigorously? 

Reviewer #1: No

Reviewer #3: Yes

4. Have the authors made all data underlying the findings in their manuscript fully available?

Reviewer #1: Yes

Reviewer #3: (No Response)

5. Is the manuscript presented in an intelligible fashion and written in standard English?

Reviewer #1: Yes

Reviewer #3: No

6. Review Comments to the Author

Reviewer #1: Thank you for the revision and detailed reply to my comments. The theoretical contribution still needs to be highlighted, issue with construct (attitude) could have been addressed with an additional study.

Reviewer #3: The paper tackles an interesting topic and I believe that it can be a relevant contribution to the literature. However, I would suggest some revisions to improve the paper.

Although I appreciate the paper, you need to be quite committed to read the whole paper and really get all the details of the study. In the current form the reader needs to go back and forth to understand the design (and hence evaluate the results) of the study.

The readability of the introduction could be improved. Moreover, many major details of the study are not introduced here, so the reader needs to read much further before getting a clear picture of the study.

Given the wide multi-disciplinary readership of Plos-One, I would suggest extending slightly the introduction to help the reader understand the focus and the relevance of the paper. The first two pages of the manuscript are at points a bit cryptic for a first-time reader. Some suggestions: the authors refer to “theory” (end of page 3) without describing them. In this point you do not need to discuss them, as they are discussed in the following sections but to introduce them. You could just add a few words (it could really be only 1-3 words) to characterise the type of theories you are referring to. Similarly, a sentence could be added to clarify in the first page, what you mean by “share results of a set of norms”. The reader at this point does not have a clear picture of what these norms may be. The term norm has different nuances in different research fields and there are “different types” of norms (e.g., descriptive, injunctive).

In section 2 ‘Consumer products as multiple dimensional percepts’ you present an interesting discussion on several product dimensions that are important to evaluate. Before the first paragraph it could be useful to add a figure (or a table if preferred) to highlight the 6 dimensions of your framework to build a conceptual map in the mind of the reader. These are well explained in Table 1, but this is at page 15!

That said, the literature review is well done and well readable.

Methods are presented in good detail, however some details could be specified further.

The authors could list in the text of the paper the 6 calibration products and discuss why they were chosen. When firstly discussing about the 150 products the authors could start to introduce what categories of products are included in the study, just to give some information to the readers. In the current form a reader needs to skim forward just to get an idea.

It is not clear if for the calibration products respondents were asked to answer only to the questions of one dimension or of all 6 dimensions. Please specify this better.

In the additional material tables, some products are in bold. I might guess (but it should be clearly stated both in the text and the tables - that should be self-explanatory) that these six products are the calibration ones. However, the bold products are not always the same, leaving me with further confusion. The authors clearly state that “From the total 150 products, the same 6 calibration products were used to start the ratings, leaving a total of 144 non-calibration products.”

The authors could consider to include in the tables in the supplementary material also a column with the number of observations on which the descriptive statistics were calculated on. This could further highlight the calibration products. Now you can only make some inference on the rough size of the sample looking at the confidence intervals. Moreover, it would be useful to specify that in those tables the mean reported is the mean across the three dimension specific questions of all respondents. The tables as they stand now are not self explanatory.

The authors also do not discuss the sampling method adopted. How were people selected (random, quota sampling, etc.)? How representative are they of North America?

Given that the authors discuss only descriptive statistics these are strongly dependent on sample composition. The authors state that the different people were assigned to the different groups randomly. Given the reduced size (i.e., around 20) of the product-dimension specific sub-samples, it would be important to discuss if respondents in these subsamples were well differentiated for socio-demographic variables (age, gender, income, education, where they live, etc,) and representative of the target population or not. Did the authors use quota sampling? And if so on what variables?

This is important to establish the inferential value of the results.

At page 19 when the authors state: “Next, we computed correlations among the six dimensions across the evaluations of all 150 products using products as the unit of analysis” I think they should specify something like “and the mean values over the product specific respondents as observed feature”, if this is what they have done!

Result description and discussion is very concise. The authors do not comment much on the specific results and do not comment at all about the products (products are many, but the authors could discuss some interesting examples) leaving much of the work of thinking about the implications of the study to the reader. It would be useful for practitioners if the authors could discuss more deeply their findings also highlighting the practical implications for applied research that would like to take into account their results. This would further highlight the relevance of the study.

The paper is written in good English, but it should be re-read carefully to fix a few typos (e.g., missing “to” in “relate to” at page 3; Bellow at page 4; “this dimensions” at page 8). The sentence “Consumer product dimensions that have been the focus of attention on research are first reviewed below and then incorporated into our normative study” at page 5 is a bit awkward.

7. PLOS authors have the option to publish the peer review history of their article (what does this mean?). If published, this will include your full peer review and any attached files.

Reviewer #1: No

Reviewer #3: No

---

## [Author Response · Author response to Decision Letter 1]

27 Jul 2020

The text below is a direct copy of the content in the uploaded file "Response to reviewers":

Dr. Claudio Soregaroli

Academic Editor

PLOS ONE

Dear Dr. Claudio Soregaroli,

We would like to thank you and the reviewers for the constructive comments in the second review round of the paper “Norms for 150 consumer products: perceived complexity, quality objectivity, material/experiential nature, perceived price, familiarity and attitude.”

We also thank you for asking the opinion of a third reviewer and for inviting us to submit a revised version of the manuscript addressing the points raised during the second review process.

Below, we address all the concerns raised by Reviewer 1 and 3 and, as suggested by you, we carefully consider the points related to the manuscript’s clarity to the reader, and hope to make the paper’s goals and contribution more clear.

We look forward to receiving your response to the new version of our revised manuscript and we thank you for your consideration and support to this work.

Kind regards, 

Filipe Loureiro

Teresa Garcia-Marques

Duane Wegener

Reviewers' comments and Authors’ responses

Reviewer #3

Comment 1: The paper tackles an interesting topic and I believe that it can be a relevant contribution to the literature. However, I would suggest some revisions to improve the paper.

Although I appreciate the paper, you need to be quite committed to read the whole paper and really get all the details of the study. In the current form the reader needs to go back and forth to understand the design (and hence evaluate the results) of the study.

The readability of the introduction could be improved. Moreover, many major details of the study are not introduced here, so the reader needs to read much further before getting a clear picture of the study.

Given the wide multi-disciplinary readership of Plos-One, I would suggest extending slightly the introduction to help the reader understand the focus and the relevance of the paper. The first two pages of the manuscript are at points a bit cryptic for a first-time reader. Some suggestions: the authors refer to “theory” (end of page 3) without describing them. In this point you do not need to discuss them, as they are discussed in the following sections but to introduce them. You could just add a few words (it could really be only 1-3 words) to characterise the type of theories you are referring to. 

Similarly, a sentence could be added to clarify in the first page, what you mean by “share results of a set of norms”. The reader at this point does not have a clear picture of what these norms may be. The term norm has different nuances in different research fields and there are “different types” of norms (e.g., descriptive, injunctive).

Response: We thank the reviewer for the comment and suggestions regarding further improving the introduction. Attending to the point raised concerning the wide multi-disciplinary readership of PlosOne, we added additional information to the introduction to help the reader understand the focus and relevance of the paper. Specifically, and as suggested, we clarify the type of theories we refer to in page 3:

Pages 3-4, lines 21-1:

“This work should contribute to a faster development of appropriate operationalizations that correspond to conceptual variables addressed by a number of theoretical approaches in the extant literature (e.g., theories of consumer perception, choice and behavior).”

We also clarify the type of norms we refer to in this work:

Page 3, lines 17-19:

“Such control can be attained either by extensive pretesting of materials or reliance on general norms published in the literature (i.e., norms for people’s shared perceptions of features specific to particular stimuli).”

Page 3, lines 20-21:

“Our goal here is to share with the scientific community a normative dataset for people’s perceptions of several consumer products regarding relevant dimensions.”

Additional information regarding the importance of these norms can also be consulted on page 4, lines 7-12:

“Our approach in the present work is the same followed in many research areas, wherein norms have been developed to allow researchers to select stimuli according to specific attributes in order to achieve appropriate experimental control (see [26]). The present normative database will support future research efforts to avoid the interference of confounding factors concerning consumer product dimensions, allowing researchers to select product stimuli in a way that controls for these features.”

Comment 2: In section 2 ‘Consumer products as multiple dimensional percepts’ you present an interesting discussion on several product dimensions that are important to evaluate. Before the first paragraph it could be useful to add a figure (or a table if preferred) to highlight the 6 dimensions of your framework to build a conceptual map in the mind of the reader. These are well explained in Table 1, but this is at page 15!

That said, the literature review is well done and well readable.

Response: We thank the reviewer for the comment and added a figure to page 5.

Comment 3: Methods are presented in good detail, however some details could be specified further. The authors could list in the text of the paper the 6 calibration products and discuss why they were chosen. 

Response: As suggested by the reviewer (besides presenting them in bold at the 

Supplementary Materials tables), the calibration products are now presented in Table 1, 

on pages 15-16. Additionally, we discuss why and how these were chosen:

Page 13, lines 10-12:

“These calibration items were presented first with the aim of providing participants a sense of the range of the stimuli to be evaluated on that dimension (e.g., [111–113]).”

Page 13, lines 14-17:

“The calibration products for each dimension (presented in Table 1 and bolded in Tables 1-6 of the Supplementary Materials) were selected based on previous research and pre-testing (e.g., perceived complexity: [22,34]; quality objectivity: [22]; material/experiential purchase: [21,114]).”

Comment 4: When firstly discussing about the 150 products the authors could start to introduce what categories of products are included in the study, just to give some information to the readers. In the current form a reader needs to skim forward just to get an idea.

Response: As suggested by the reviewer, when presenting our stimuli, we added some 

brief information about the categories of products included in the normative dataset.

Page 13, lines 4-6:

“The full list can be consulted in the Supplementary Materials and includes products such as food, clothing, electronics, household products, experiences and services.”

Comment 5: It is not clear if for the calibration products respondents were asked to answer only to the questions of one dimension or of all 6 dimensions. Please specify this better.

Response: We thank the reviewer for pointing this out. In light of this comment we tried to specify this better.

Page 13, lines 7-10:

“To prevent task fatigue and demotivation, each participant evaluated 54 products from the total list of 150 products for one of the six dimensions described. From those 54 products, participants first evaluated a set of 6 calibration products that spanned the range of the single dimension evaluated.”

Comment 6: In the additional material tables, some products are in bold. I might guess (but it should be clearly stated both in the text and the tables - that should be self-explanatory) that these six products are the calibration ones. However, the bold products are not always the same, leaving me with further confusion. The authors clearly state that “From the total 150 products, the same 6 calibration products were used to start the ratings, leaving a total of 144 non-calibration products.”

Response: The indication that the products in bold in the Supplementary Materials correspond to the calibration products was present in the previous version of the manuscript (previous version: p. 13, lines 12-13: “The calibration products for each dimension (in bold on Tables 1-6 of the Supplementary Materials) were selected based on previous research”). This sentence was adjusted in the current version of the manuscript in order to respond to comment 3:

Page 13, lines 14-17:

“The calibration products for each dimension (presented in Table 1 and bolded in Tables 1-6 of the Supplementary Materials) were selected based on previous research and pre-testing (e.g., perceived complexity: [22,34]; quality objectivity: [22]; material/experiential purchase: [21,114]).”

Additionally, this information was also added, as a note, to the tables in the Supplementary Materials. 

The reviewer is correct in observing that the calibration products are not the same, as a product that represented the low end of one dimension, for example, might not also represent the low end of another dimension. The calibration products were chosen to play that role for each given dimension, not across all dimensions at once. The variation in calibration products across the assessed dimensions is now made clear in the current version of the manuscript. To avoid the confusion possibly induced by the footnote on page 13, we made some changes:

Page 13, footnote:

“From the total of 150 products, 6 calibration products were chosen to represent the range of values for that dimension and were used to start the ratings, leaving a total of 144 non-calibration products for each dimension.”

Comment 7: The authors could consider to include in the tables in the supplementary material also a column with the number of observations on which the descriptive statistics were calculated on. This could further highlight the calibration products. Now you can only make some inference on the rough size of the sample looking at the confidence intervals. Moreover, it would be useful to specify that in those tables the mean reported is the mean across the three dimension specific questions of all respondents. The tables as they stand now are not self explanatory.

Response: We thank the reviewer for the suggestion. In line with that, we included in the tables in the Supplementary Materials another column with the number of observation for each product. Additionally, we specify that the mean reported is the mean across the three dimension-specific items.

Comment 8: The authors also do not discuss the sampling method adopted. How were people selected (random, quota sampling, etc.)? How representative are they of North America?

Given that the authors discuss only descriptive statistics these are strongly dependent on sample composition. The authors state that the different people were assigned to the different groups randomly. Given the reduced size (i.e., around 20) of the product-dimension specific sub-samples, it would be important to discuss if respondents in these subsamples were well differentiated for socio-demographic variables (age, gender, income, education, where they live, etc,) and representative of the target population or not. Did the authors use quota sampling? And if so on what variables?

This is important to establish the inferential value of the results.

Response: Participants were selected form an eligible US population size of 32,842 active participants on Prolific. US citizens are well represented in this platform and correspond to the second best represented country on the platform (after UK, with 40,297 active participants). We do not have information on participants’ socio-demographic variables other than age and gender. Cross-stratification options on the platform are only possible on the following factors: sex, age, ethnicity (Simplified US Census).

While quota sampling would indeed be useful to establish a representative sample of the population of North America, it was not an explicit goal of this work to accomplish such a representativeness, but instead, to study people’s general perceptions of certain stimuli according to specific features (as it is the case with other normative studies). The overall sample of participants in this study is, nevertheless, well distributed across gender, and the median age is close to that of the US general population. Additionally, the confidence intervals reported across all dimensions suggest good uniform reliability.

Nevertheless, we do believe that generalizations should be made with caution and recommend that cross-validation of these norms are performed, as it is the case with other normative datasets (as stated on page 20, lines 17-19).

Comment 9: At page 19 when the authors state: “Next, we computed correlations among the six dimensions across the evaluations of all 150 products using products as the unit of analysis” I think they should specify something like “and the mean values over the product specific respondents as observed feature”, if this is what they have done!

Response: In line with the reviewer’s comment we added the information:

Page 19, lines 15-17:

“Next, we computed correlations among the six dimensions across the evaluations of all 150 products, using products as the unit of analysis (i.e., correlating the mean values for each product’s features).”

Comment 10: Result description and discussion is very concise. The authors do not comment much on the specific results and do not comment at all about the products (products are many, but the authors could discuss some interesting examples) leaving much of the work of thinking about the implications of the study to the reader. It would be useful for practitioners if the authors could discuss more deeply their findings also highlighting the practical implications for applied research that would like to take into account their results. This would further highlight the relevance of the study.

Response: In line with the goals of the paper, we highlight the most important specific results (product-related and dimensions-related) along the three reported tables and presented graphic. Specifically, we start by analyzing the evaluations regarding each dimension across all products (Table 2), before we present a list of the most extreme positive and negatively evaluated products for each dimension (Table 3), emphasizing their overall variation across the range of the dimensions evaluated (also supported by Figure 2). Finally, the analysis of the obtained correlations between the six dimensions across all products (Table 4) highlights the practical implications for applied research and suggests the importance of controlling for these product features, by evidencing several significant correlations among the assessed dimensions, suggesting that these are confounded across products. 

As mentioned by the reviewer, the products in this dataset are many, but we do believe that the biggest implications of the obtained results are well explored (and fulfill the goals of this paper), introducing the reader to 1) further explore the individual results regarding the 150 products and 2) be able to select consumer products according to specific attributes and facilitate researchers’ choices aimed at achieving appropriate experimental control.

Comment 11: The paper is written in good English, but it should be re-read carefully to fix a few typos (e.g., missing “to” in “relate to” at page 3; Bellow at page 4; “this dimensions” at page 8). The sentence “Consumer product dimensions that have been the focus of attention on research are first reviewed below and then incorporated into our normative study” at page 5 is a bit awkward.

Response: We thank the reviewer for pointing out these potential typos. We believe that the first of these (on page 3) is phrased correctly (i.e., because it uses “to which participants can easily relate” it would not seem correct to add “to” after “relate” – the added “to” would be redundant with “to which,” and the “to which” phrasing is preferred). The other sentence referenced by the reviewer was changed to:

Page 5, lines 13-14:

“We first review some of the consumer product dimensions that have been the focus of research attention (see Figure 1) before operationalizing them in our normative study.”

In addition to paying attention to these particular sentences, we made further revisions wherever any other typos were detected.

Reviewer #1: 

Comment 1: Thank you for the revision and detailed reply to my comments. The theoretical contribution still needs to be highlighted, issue with construct (attitude) could have been addressed with an additional study.

Response: We thank the reviewer for the comment, and hope to have contributed to make this work’s goals (as well as its theoretical contribution) more clear with the changes implemented in this second review round. We don’t believe that, within the purposes of this normative dataset, there is an issue per se with the attitude construct or with how it was operationalized (much in line with our previous response to this point in the last review round). We hence believe that an additional study addressing an alternative operationalization to this construct would not significantly benefit the aims and goals of the current work.

---

## [Decision Letter · Decision Letter 2]

26 Aug 2020

Norms for 150 consumer products: perceived complexity, quality objectivity, material/experiential nature, perceived price, familiarity and attitude.

PONE-D-20-00355R2

Dear Dr. Loureiro,

We’re pleased to inform you that your manuscript has been judged scientifically suitable for publication and will be formally accepted for publication once it meets all outstanding technical requirements.

Kind regards,

Claudio Soregaroli

Academic Editor

PLOS ONE

Reviewers' comments:

Reviewer's Responses to Questions

**Comments to the Author**

1. If the authors have adequately addressed your comments raised in a previous round of review and you feel that this manuscript is now acceptable for publication, you may indicate that here to bypass the “Comments to the Author” section, enter your conflict of interest statement in the “Confidential to Editor” section, and submit your "Accept" recommendation.

Reviewer #3: (No Response)

2. Is the manuscript technically sound, and do the data support the conclusions?

Reviewer #3: Yes

3. Has the statistical analysis been performed appropriately and rigorously? 

Reviewer #3: Yes

4. Have the authors made all data underlying the findings in their manuscript fully available?

Reviewer #3: (No Response)

5. Is the manuscript presented in an intelligible fashion and written in standard English?

Reviewer #3: Yes

6. Review Comments to the Author

Reviewer #3: The authors have addresses most comments and the paper has improved.

7. PLOS authors have the option to publish the peer review history of their article (what does this mean?). If published, this will include your full peer review and any attached files.

Reviewer #3: **Yes: **Elena Claire Ricci

---

## [Editor Report · Acceptance letter]

11 Sep 2020

PONE-D-20-00355R2 

Norms for 150 consumer products: perceived complexity, quality objectivity, material/experiential nature, perceived price, familiarity and attitude. 

Dear Dr. Loureiro:

I'm pleased to inform you that your manuscript has been deemed suitable for publication in PLOS ONE. Congratulations! Your manuscript is now with our production department. 

Kind regards, 

on behalf of

Dr. Claudio Soregaroli 

Academic Editor

PLOS ONE